# Growth dynamics and amorphous-to-crystalline phase transformation in natural nacre

L. M. Otter [1] ✉, K. Eder [2], M. R. Kilburn[3], L. Yang[2,4], P. O'Reilly[5], D. B. Nowak[5], J. M. Cairney [2] & D. E. Jacob [1]

Biominerals, such as nacreous bivalve shells, are important archives of environmental information. Most marine calcifiers form their shells from amorphous calcium carbonate, hypothesised to occur via particle attachment and stepwise crystallisation of metastable precursor phases. However, the mechanism of this transformation, including the incorporation of trace elements used for environmental reconstructions, are poorly constrained. Here, using shells of the Mediterranean mussel, we explore the formation of nacre from the meso- to the atomic scale. We use a combination of strontium pulse-chase labelling experiments in aquaculture and correlated micro- to sub-nanoscale analysis to show that nacre grows in a dynamic two-step process with extensional and space-filling growth components. Furthermore, we show that nacre crystallizes via localised dissolution and reprecipitation within nanogranules. Our findings elucidate how stepwise crystallization pathways affect trace element incorporation in natural biominerals, while preserving their intricate hierarchical ultrastructure.

A growing body of evidence indicates that calcifying organisms form their hard parts (e.g., shells) from amorphous calcium carbonate (ACC), which later transforms via a stepwise crystallisation of intermediate metastable phases rather than by direct precipitation of aragonite and calcite from seawater[1,2]. Observations of ACC in natural bio-carbonate systems such as echinoid spicules[3], foraminifera shells[4] and mollusc nacre[4,5] were among the main drivers of the paradigm change in crystallisation theory from classical monomer-by-monomer processes towards non-classical crystallisation pathways[6]. In fact, crystallisation via particle attachment and stepwise transformation of metastable precursor phases are now accepted as crystallisation models for many different systems[1–8].

Bio-carbonates are important archives for both environmental reconstructions and forward-modelling of future environmental and climatic conditions, including sea-surface temperatures, ocean pH and salinity[9–11]. Presently, these reconstructions are based on the incorrect assumption that bio-calcite and -aragonite form directly via precipitation from seawater, while more realistic modelling of uptake and distribution of elements and isotopes through a stepwise crystallisation pathway via ACC is still lacking.

A critical knowledge gap in working towards a new model for element and isotope partitioning in natural bio-carbonates is the lack of insight into the mechanisms by which the different intermediate calcium carbonate phases control the chemical composition of the final stable phase. Generally, the composition of crystalline phases is governed by their distinct crystal chemistry and can be quantified using equilibrium partition coefficients, defined as the concentration ratio of a specific element between two phases[12]. As a general principle, amorphous materials lack defined partition coefficients and are thus less selective with respect to trace element incorporation compared to

[1]Research School of Earth Sciences, Australian National University, Canberra, ACT 2601, Australia. [2]Australian Centre for Microscopy and Microanalysis, The University of Sydney, Sydney, NSW 2006, Australia. [3]Centre for Microscopy Characterisation and Analysis, University of Western Australia, Perth, WA 6009, Australia. [4]School of Civil & Environmental Engineering, University of Technology Sydney, Ultimo, NSW 2007, Australia. [5]Molecular Vista Inc., 6840 Via Del Oro, Suite 110, San Jose, CA 95119, USA. ✉e-mail: Laura.Otter@anu.edu.au

their crystalline counterparts. In the case of ACC, stepwise crystallisation occurs either via dissolution and reprecipitation, solid-solid transformation or a combination thereof[13]. While dissolution and reprecipitation mechanisms are governed by equilibrium partition coefficients[12,14,15], chemical re-distribution during solid-solid transformation can deviate significantly from equilibrium partitioning and instead preserves the original chemical composition of ACC in the crystalline end-product. Thus, either transformation pathway has a significantly different effect on element partitioning into the final crystalline phase.

In the laboratory, ACC typically transforms via a dissolution and reprecipitation mechanism[13–17]. This pathway is usually associated with significant morphological changes to the texture of the final crystalline phase[13–18]. However, the preservation of intricate hierarchical structural details in natural systems during stepwise crystallisation seems to contradict the findings in the laboratory and instead support either solid-solid transformation, or very localised dissolution-reprecipitation processes.

We aim here to elucidate the transformation mechanisms from ACC to aragonite during bivalve nacre growth using living Mediterranean mussel shells (*Mytilus galloprovincialis)* and a combination of Sr pulse-chase labelling experiments with correlated micro- to sub-nanometre analysis. *M. galloprovincialis* shells form via a stepwise crystallisation pathway from ACC to aragonite[4,19,20]. Our pulse-chase labelling aquaculture experiments with living bivalves (see methods) produced shells with well-defined bands of Sr-enriched CaCO₃ that are visible under the electron and ion beams and serve as timestamps of shell growth[21]. Aquaculture-based pulse-chase labelling experiments employing trace elements, such as strontium and magnesium, have significantly furthered our understanding of the growth dynamics in various corals and sea urchins[22,23], yet, similar applications to bivalves are rare[21]. Correlated analysis using Nanoscale Secondary Ion Mass Spectrometry (NanoSIMS), Atom Probe Tomography (APT) and nanoscale Photo-induced Force Microscopy (PiFM) enabled stepwise spatially downscaled visualisation of the time-stamped nacre portions from the meso- to the atomic scale. We show that nacre forms via a two-stepped process of extensional followed by space-filling growth, a process that has long been suggested[24–26] but is directly observed here via the Sr-labelled growth sequence. Most importantly, we demonstrate that the transformation of ACC to aragonite in natural nacre occurs via dissolution and reprecipitation within individual

nanogranules, thus elucidating the mechanism behind the preservation of the hierarchical nanostructures in natural biominerals upon stepwise crystallisation.

## Results

### Nacre growth visualised by strontium pulse-chase labelling

The aragonitic nacreous shell layer in *M. galloprovincialis* is the innermost of two calcified layers (Fig. 1a, b) and contains up to 3 wt.%[27] of various organic molecules, which provide an architectural framework[24], and control mineralisation[28]. CaCO₃ and organics are secreted by the epithelial cells of the organism's mantle tissue which covers the entire inner surface of the shell. Mature nacre consists of ca. 10–20 μm wide and 0.5 μm thick polygonal aragonite tablets which are individually enveloped in organic sheets of ca. 30 nm thickness[27,29,30]. In addition, significant amounts of nanometre-sized organic inclusions are found in each nacre tablet[27,31]. Nacre consists of space-filling nanogranules of ca. 20–150 nm in size[8,32–37]. New layers of nacre are added via mineral bridges through pores in the organic sheets[27,29,38,39] and mature to aragonite[1,4,5,20].

Bivalve shells grown during our pulse-chase labelling aquaculture experiment have an average Sr concentration difference of 10,434 μg/g between labelled and unlabelled nacre portions (see methods, Supplementary Tables 1 and 2) which translate into Sr-labelled shell portions appearing as lighter grey layers in-between the darker grey portions of unlabelled shell in Backscattered Electron (BSE) images (Fig. 1c, Supplementary Fig. 1). Natural variations in growth rates result in different widths of Sr-labelled shell portions within and between individual bivalve shells[21]. Here, we selected specimens for further high-resolution analysis that had the highest growth rates to achieve best spatial resolution with our methods (Supplementary Table 1).

At the micron- to nanoscale, NanoSIMS isotope ratio distribution maps of the nacre in Sr-labelled shells show a stepped growth pattern along the inner shell edge (Fig. 2a, Supplementary Figs. 2 and 3) with regions of distinctly high Sr/Ca- ratios (warm colours in Fig. 2) intercalated with areas of low Sr/Ca ratios (blue colours). Abrupt high-angle changes in colour from magenta to blue and vice versa in the maps mark the transitions from Sr-labelled to unlabelled tablet portions (Fig. 2b, white arrowheads). The transitions between high and low Sr/Ca areas cut through individual nacre tablets (Fig. 2b). Green to yellow pixels along these transitions reflect intermediate Sr/Ca ratios, likely

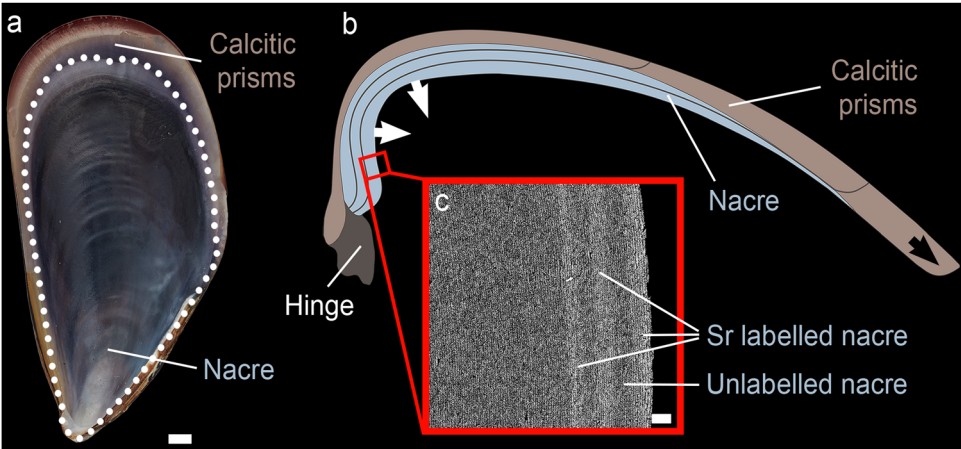

**Fig. 1 | Nacre growth visualised by Sr pulse-chase labelling. a** Photo of the inner shell surface of a *M. galloprovincialis* shell showing nacre constituting most of the inner shell surface inside the white dotted line and prismatic calcite outside the line. **b** Schematic cross-section of a shell. The shell grows in length by extending the outer calcite layer along the ventral margin (black arrow). The inner nacreous shell layer extends by thickening the inner shell surface (two white arrows). **c** SEM-BSE image of a polished shell cross-section (specimen M2S2R) showing Sr-labelled nacre layers as three light grey bands tracing parallel to the inner shell edge, while shell portions grown in normal seawater composition are shown in darker grey. In this specific example, the nacre layer grew 25 μm over the experimental period of 46 days marked by the start of the innermost Sr label. Scale bars are 1 mm (**a**) and 5 μm (**c**).

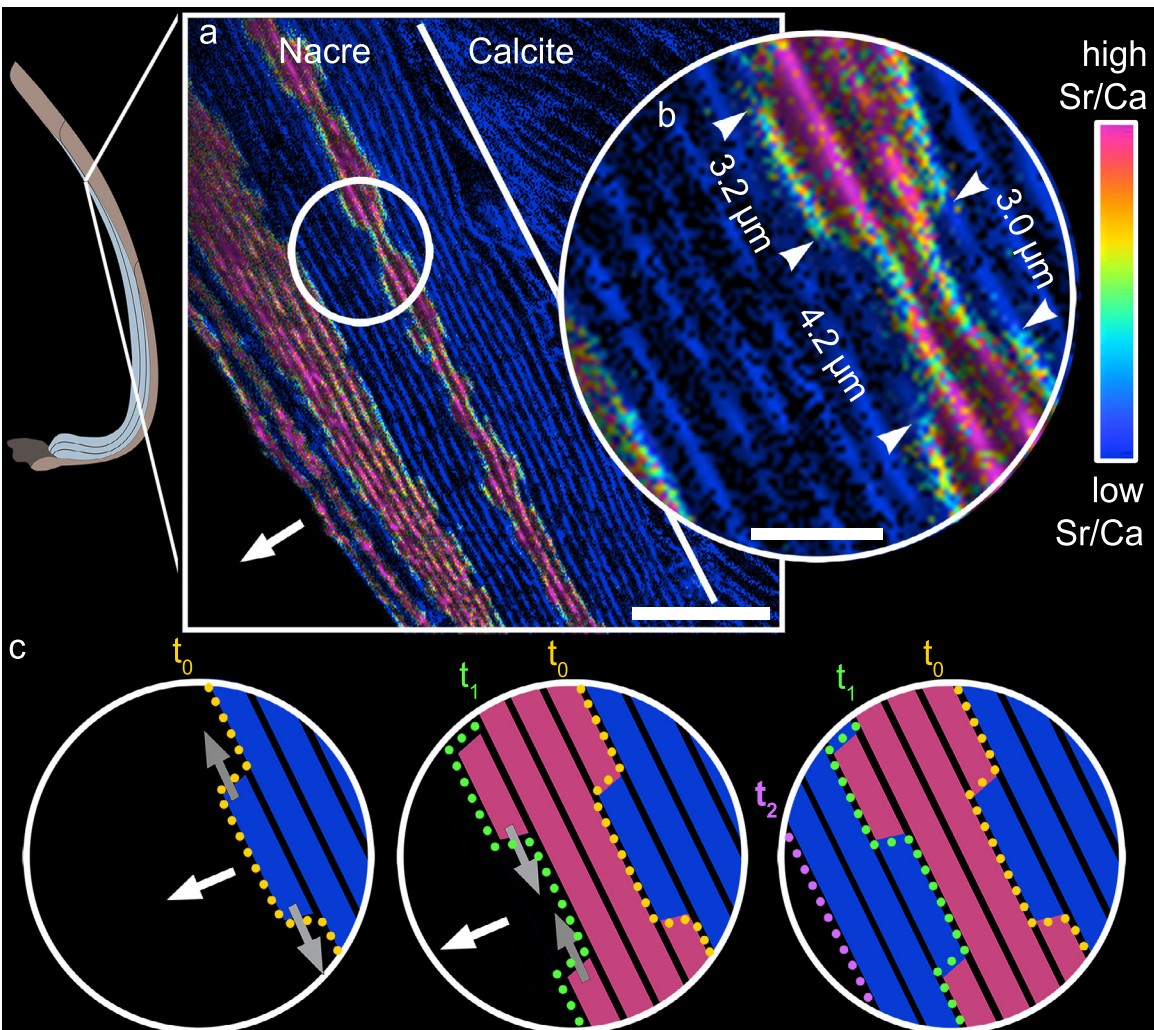

**Fig. 2 | NanoSIMS imaging of nacre growth processes visualized via Sr pulse-chase labelling. a** $^{88}Sr/^{40}Ca$ ratio map sized $50 \times 50$ μm showing four Sr labels (magenta) and unlabelled nacre (blue) parallel to the nacre growth front (specimen: M1S1L). The combined unlabelled and Sr-labelled nacre layers have average Sr concentrations of 2,100 μg/g and 12,500 μg/g, respectively (Supplementary Table 2). The labelled nacre layers exhibit a stepped growth pattern of intercalated nacre tablets formed with normal Sr concentration (blue) and during Sr-labelling (magenta). **b** Magnified region of interest (circled in a) shows sharp (i.e., within 100 nm) near-vertical transitions, marked by white arrowheads, between labelled and unlabelled portions of nacre lamellae. The sizes of neighbouring Sr-labelled and unlabelled lamellae portions ranging between 3.0 and 4.2 μm indicate that these portions represent changes in Sr/Ca ratios within individual tablets as the total length of a nacre tablet is 10–20 μm. **c** Time-resolved schematic representation of the two components of nacre growth processes (area corresponds to that shown in b): extensional nacre growth across layers normal to the organic inter-lamellar sheets (white arrows) and space-filling nacre growth of individual tablets parallel with the interlamellar sheet within individual nacre layers (grey arrows). Growth between $t_0$ (yellow dotted line) and $t_1$ (green dotted line) was achieved within 3 days and between $t_1$ and $t_2$ (purple dotted line) within the following 6 days (see Supplementary Movie 1 for a full animation of the growth sequence). This time-resolved illustration of nacre growth demonstrates that extensional growth, thickening the shell, is followed by space-filling growth of separate individual nacre tablets. For additional NanoSIMS maps see Supplementary Figs. 2 and 3. Scale bar is 10 μm (**a**) and 3 μm (**b**).

representing concentration mixtures beyond the spatial resolution of individual pixels (see methods).

The stepped growth pattern visualised via the timestamps created by Sr labelling shows that nacre grows along two directions at sub-micrometre length scales that we refer to here as extensional and space-filling growth: extensional growth forms new nacre layers along the inner shell surface and progresses quasi-simultaneously at multiple locations of different tablets, normal to the organic interlamellar sheets (in Fig. 2c, white arrows). This is followed by space-filling nacre growth parallel to the organic interlamellar sheets (Fig. 2c, grey arrows; Supplementary Movie 1) which proceeds via incremental additions of $CaCO_3$ nanogranules to the growing nacre tablets. This two-stepped mode of growth, consisting of an extensional followed by a space-filling growth component, was previously observed for corals[40] but has so far only been inferred for mollusc shells[24,25].

## Strontium heterogeneity in nanogranules

We used Atom Probe Tomography (APT) to explore atomic-scale element distribution patterns in Sr-labelled nacre and subsequently correlate these analyses with Photo-induced Force Microscopy (PiFM). Regions of interest in the Sr-enriched shell portion were prepared using the interlamellar organic sheet as a point of reference for atom probe-tip preparation during BSE imaging and simultaneous Focussed Ion Beam (FIB) cutting (see methods and Supplementary Fig. 4). We analysed three different areas in one atom probe tip (Supplementary Fig. 5). The colour-coded 3D APT reconstructions (Fig. 3a–f, Supplementary Figs. 6–9) show the Ca-rich mineral phase of the nacre tablets in grey to the left, while high concentrations of C (magenta) and H (yellow) identify the organic interlamellar sheet to the right (Fig. 3a–b). A 3.8 nm wide area of nacre is significantly enriched in Sr (Fig. 3c–d) and an atom probe concentration profile (Fig. 4) further demonstrates

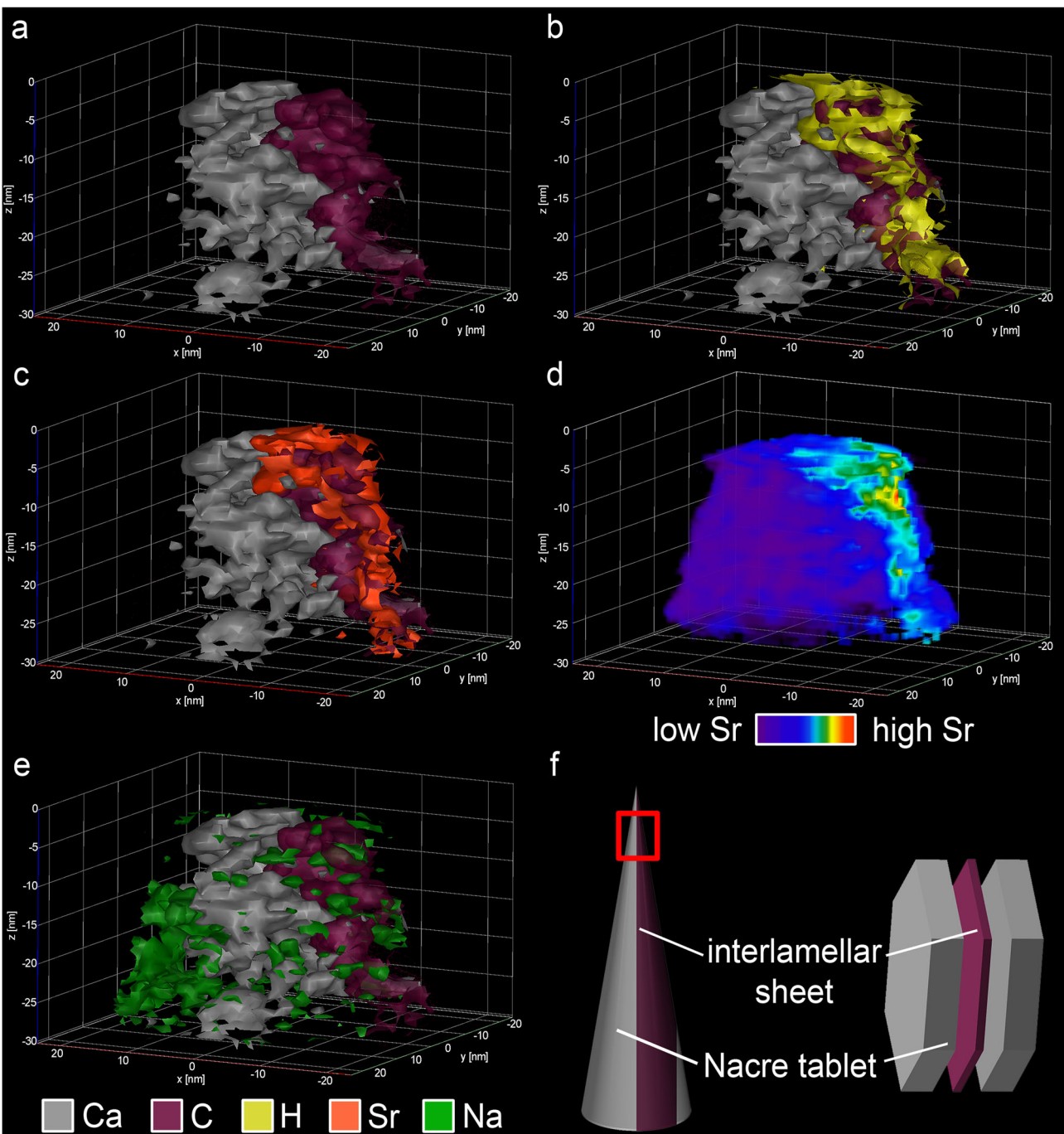

**Fig. 3 | 3D APT reconstruction showing parts of the nacre tablet and the organic interlamellar sheet in Sr-labelled nacre (specimen: M2S2R). a** Iso-concentration surfaces of Ca (50 at%, grey) depicting the outer area of a nacre tablet and elevated C (5.1 at%, magenta) identifies the organic interlamellar sheet. **b** Iso-concentration surfaces of H (10 at%, yellow) co-located with high C concentrations in the organic interlamellar sheet. **c** The iso-concentration surface of Sr (20 at%, red) shows a distinct Sr-rich area situated between Sr-poor aragonite within the mineral fraction. **d** Volume rendering showing the distribution of Sr across the volume of the reconstruction highlighting the Sr-enriched area (red to yellow) in the nacre tablet. **e** Iso-concentration surfaces of Na (8 at%, green) forms a distinct cluster within the mineral phase next to the Sr-rich area. **f** Schematic representation of the APT reconstruction (red box) and its relative position within the tip as well as within the nacreous architecture more generally.

that the location of the Sr enrichment is in the mineral part but not in the organic sheet. Other reconstructed regions of the tip (Supplementary Figs. 6–9) show comparable results with significantly Sr-enriched areas of similar thicknesses (4.5 nm and 5.0 nm) in the mineral part of nacre. Sodium concentrations are enriched both close to and inside the organic sheet (Figs. 3e and 4, Supplementary Figs. 6–9). Similar Na enrichment at the mineral-organic interface were reported for calcitic foraminifera[41].

The compositional information derived from atom probe analysis of the nacre tablet was correlated with Photo-induced Force Microscopy (PiFM). PiFM is a nanoscale AFM-based technique that raster-scans the sample surface at high spatial resolution. It measures the force created in a very small interaction volume between the AFM cantilever tip and the sample surface, while a laser sweeps through the infrared spectrum causing absorption-specific variations in the recorded force[42]. PiFM absorption spectra agree with those produced by

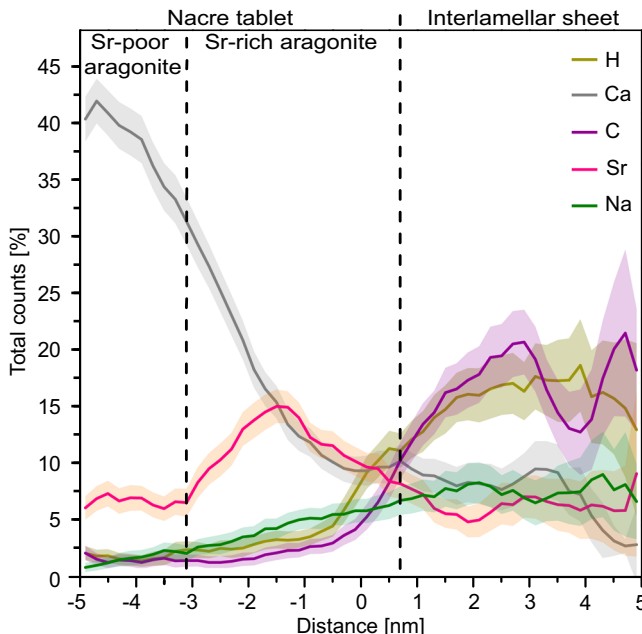

**Fig. 4 | Compositional profile through the APT reconstruction across the nacre tablet.** The qualitative profile across the APT reconstruction shown in Fig. 3. The plot is divided by black dashed lines into three sections: Sr-poor aragonite, Sr-rich aragonite and interlamellar sheet. The transition from Sr-poor to Sr-rich aragonite is defined by a significant increase in Sr, while the transition from Sr-rich aragonite to the organic interlamellar sheet is defined as the point of intersection between the Ca (grey) and the C (magenta) signals (see methods). The Sr-poor aragonite is characterised by high Ca concentrations indicative of $CaCO_3$, while the Sr-rich area is defined by high Sr (red) at intermediate Ca and the interlamellar sheet by high C and H (yellow) counts. The Sr-enriched area appears within 4 nm of the mineralised nacre tablet adjacent to the organic sheet. The abundance of Na (green) increases steadily from the tablet into the organic sheet. Shaded envelopes depict the first standard deviations.

traditional FTIR spectroscopy[43] but are obtained at a higher spectral and spatial resolution, which leads to some differences in absorption band appearance[44] particularly for materials that are heterogeneous at the nanoscale.

Here, PiFM was used to acquire phase distribution maps at a spatial resolution similar to that of the APT reconstructions, but at a larger field of view that provide more context. We obtained an overview image sized $25 \times 25\,\mu m$ of the Sr-labelled and unlabelled shell portions (Supplementary Fig. 10a, b) before obtaining a 0.5 by $0.5\,\mu m$ map from within the labelled area (Fig. 5) in Hyperspectral PiFM IR Imaging (HyPIR) mode. This mode produces a full range spectrum in each pixel of the map, such as the representative spectrum shown in Fig. 5a, which allows us to extract individual wavenumber-specific maps for aragonite (Fig. 5b), strontianite (Fig. 5c) and the organic moiety from the same region (Fig. 5c). The band at $1482\,cm^{-1}$ is the characteristic main band (asymmetric stretching) of aragonite[45] and the band at $1658\,cm^{-1}$ identifies the C=O stretching vibrations of the amide I functional group[46]. The organic phase is visible throughout the organic interlamellar sheet and as part of the intracrystalline organic matrix inside the nacre tablet. The weaker absorption band at $1446\,cm^{-1}$, located adjacent to the aragonite main band, is assigned to strontianite. Notably, this band is only present in spectra from the Sr-labelled portions of nacre (Supplementary Fig. 10c). Its assignment to strontianite is verified by deconvolving the broad main carbonate band region (i.e., from 1350 to $1650\,cm^{-1}$) of unlabelled and Sr-labelled nacre (Supplementary Fig. 11a, b) using a best quality fit Voigt model. The Voigt deconvolution of the main carbonate absorption bands for both unlabelled and Sr-labelled nacre (Supplementary Fig. 11a, b)

shows the $1482\,cm^{-1}$ peak centre associated with aragonite. The deconvolution of the Sr-labelled nacre produced two additional peaks centred at $1472\,cm^{-1}$ and $1446\,cm^{-1}$ (Supplementary Fig. 11b) that coincide to those obtained from a synthetic strontianite reference material (Supplementary Fig. 11c, see methods for details).

The AFM phase contrast image (Fig. 5b), simultaneously collected with the HyPIR map, displays the characteristic space-filling nanogranular texture of nacre with granule diameters ranging between 20 and 100 nm that are in agreement with previous studies[8,32,34]. Comparison with the PiFM phase distribution maps for aragonite (Fig. 5c) and at $1446\,cm^{-1}$ (Fig. 5d) assigned to strontianite shows that there are Sr-rich and Sr-poor aragonitic areas in each nanogranule and that these areas do not appear to straddle the contours of the nanogranules. This confinement within the boundaries of individual nanogranules is explained by the fact that in nacre, each nanogranule is sheathed by organics[8,35–37,47]. Comparable results to those shown in Fig. 5 were obtained in an adjacent area to the one shown here (Supplementary Fig. 12).

Applying principal component analysis and Multivariate Curve Resolution (PCA/MCR, see methods for details) to the raw spectra in the HyPIR map shown in Fig. 5 also shows the confinement of strontianite to individual nanogranules in the nacre sample. PCA/MCR component 1 is defined by an intense peak at $1482\,cm^{-1}$ and PCA/MCR component 2 shows its most intense peak at $1472\,cm^{-1}$ (Fig. 6a). PiFM reference spectra for pure aragonite and synthetic strontianite show that the main peak of component 1 can be assigned to the aragonite main band, while the main peak of component 2 exhibits an intermediate position between the main bands of aragonite and strontianite. This suggests that component 2 is a mixture of strontianite and aragonite beyond the 5 nm spatial resolution of the PiFM. Furthermore, the absorption bands for pure aragonite and Sr-rich aragonite agree well with those identified previously in the Voigt deconvolution (Supplementary Fig. 11). We, therefore, refer to it as Sr-rich aragonite. This is in good agreement with our APT data that show a co-location of Sr and Ca in the Sr-enriched areas.

The resulting component maps (Fig. 6b–d, Supplementary Fig. 13) verify our observations and show the within-granule heterogeneity of aragonite areas (Fig. 6b, blue) and Sr-rich aragonite (Fig. 6c, magenta). Component intensity profiles across individual nanogranules in different areas of the nacre tablet (Fig. 6e) show a systematic pattern of nanogranules consisting of Sr-poorer aragonite cores and Sr-rich aragonite towards their contours, which is in good agreement with our APT results, albeit the latter probed a much smaller area (Figs. 3 and 4 Supplementary Figs. 6–9).

## Discussion

Strontium pulse-chase labelling of bivalve shells produces timed snapshots of different stages of nacre growth, which provide direct evidence for a two-step growth process of nacre that has been inferred as early as in the 1960's[24]: Extensional nacre growth occurs along the local growth direction of the shell normal to the organic interlamellar sheets (Fig. 2c, white arrows) and is complemented by space-filling growth of nacre tablets parallel to these sheets (Fig. 2c, grey arrows). NanoSIMS maps of the Sr-labelled nacre (Fig. 2 Supplementary Fig. 2, Supplementary Fig. 3) indicate that extensional growth of nacre lamellae occurs simultaneously at multiple growth centres along the organic interlamellar sheet, forming the centres of new nacre tablets. This observation supports previous studies indicating that extensional growth proceeds via mineral bridges through pores in the organic interlamellar sheets[38,39] often associated with screw dislocation[29]. When shells are studied in cross-section, the interplay of both modes of growth result in a stepped pattern of adjacent Sr-rich (i.e., Sr-labelled) and Sr-poor (i.e., unlabelled) portions within individual nacre tablets (Fig. 2b, Supplementary Figs. 2 and 3 and Supplementary Movie 1).

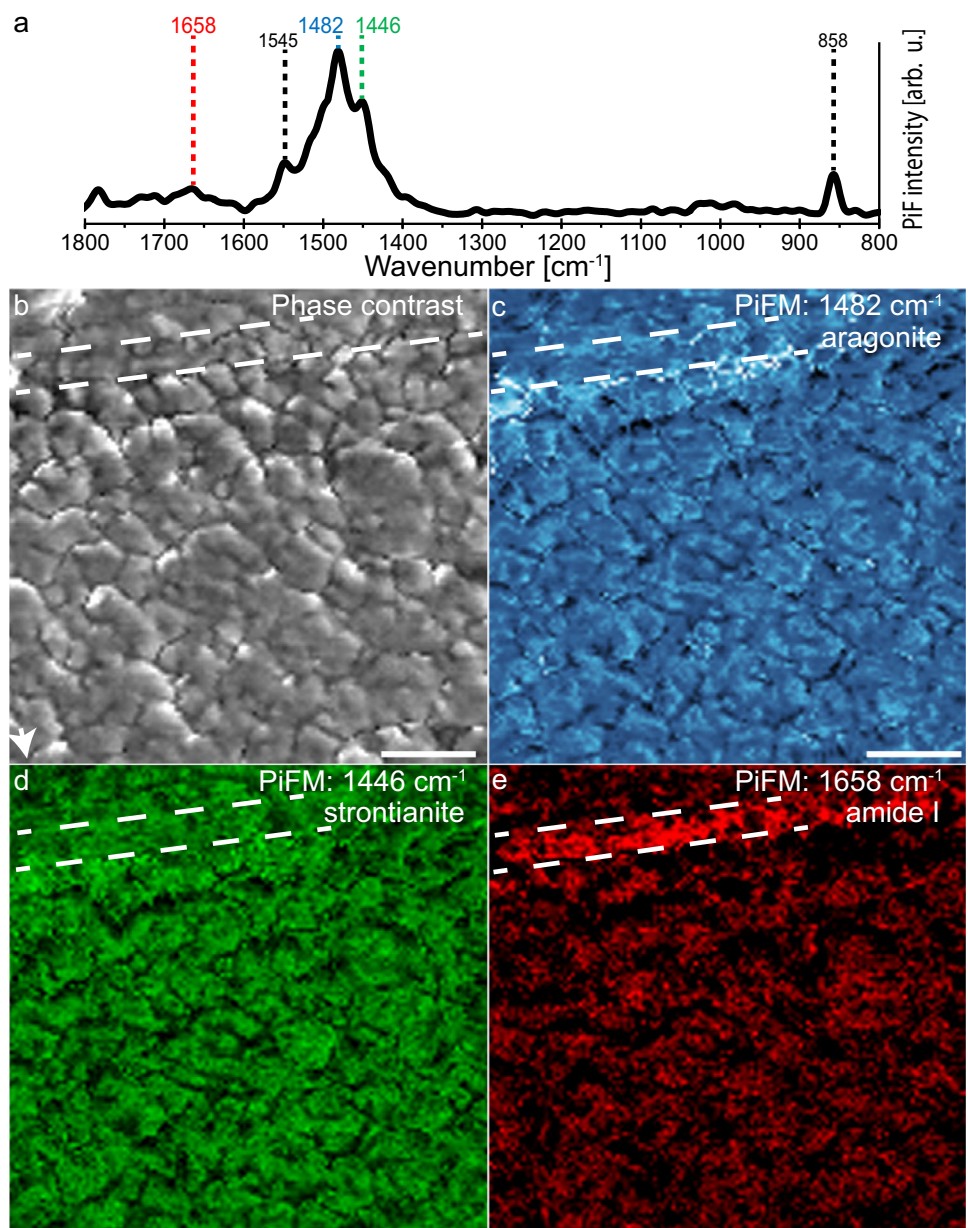

**Fig. 5 | PiFM analysis revealing heterogeneous, nanoscale phase distributions in Sr-labelled nacre. a** Representative PiFM spectrum showing the absorption bands chosen for extracting the colour-coded PiFM phase distribution maps at 1482, 1446 and 1658 cm⁻¹ as well as additional bands characteristic of aragonite (black font). **b** The 0.5 × 0.5 μm AFM phase contrast map was simultaneously acquired with the PiFM phase distribution maps obtained from Sr-labelled nacre (specimen: M2S2R) with the interlamellar organic sheet (within white dashed lines) running nearly horizontally through the mapped region. Lighter and darker colour in the phase contrast map highlight the space-filling nanogranular texture; the white arrow points towards the direction of extensional nacre growth (i.e., the inner shell surface). **c** Shows the distribution of the main aragonite absorption band mapped at 1482 cm⁻¹ that highlights the nanogranules, **d** map at 1446 cm⁻¹, which is a small absorption band seen in the spectra of Supplementary Fig. 10 that accentuates distinct portions of nanogranules and **e** distribution of the proteinaceous organic phases mapped using the amide I band at 1658 cm⁻¹ showing a strong enrichment in the organic sheet and as part of the intracrystalline organic matrix inside the nacre tablet. All four maps were acquired simultaneously in Hyperspectral PiFM IR Imaging (HyPIR) mode and have a pixel resolution of 5 nm. Comparable results were obtained in an adjacent area to the one shown here (Supplementary Fig. 12). Scale bars are 100 nm.

Individual aragonitic nanogranules, the basic structural units of nacre tablets, are shown to exhibit heterogeneous Sr concentration at nanometre resolution: They display distinct Sr-enriched areas mostly towards the contours of the nanogranules and distinct Sr-poor areas mostly towards the granule centres. Sr enrichments are clearly identified through significantly higher counts in the APT analyses (Figs. 3 and 4, Supplementary Figs. 6–9) and through the presence and intensity of an absorption band indicative of strontianite in the PiFM spectra (Figs. 5a and 6a, Supplementary Fig. 11).

Heterogeneity of similar dimensions, including in nacre nanogranules have been previously reported in the literature[19,34] where they were identified by phase composition (i.e., the presence or absence of ACC) rather than based on their chemical composition: Nassif et al.[19] reported similarly sized, 3–5 nm wide ACC layers coating the nacre tablets in *Haliotis laevigata*. In the shell of *Phorcus turbinatus* and at spatial resolution of individual nanogranules, Macias-Sanchez et al.[34] observed 5–10 nm wide ACC coatings along the exteriors of crystalline aragonite nanogranules (referred to as nanoglobules). Both studies[19,34] argue that these ACC-rich areas represent vestiges of the non-classical

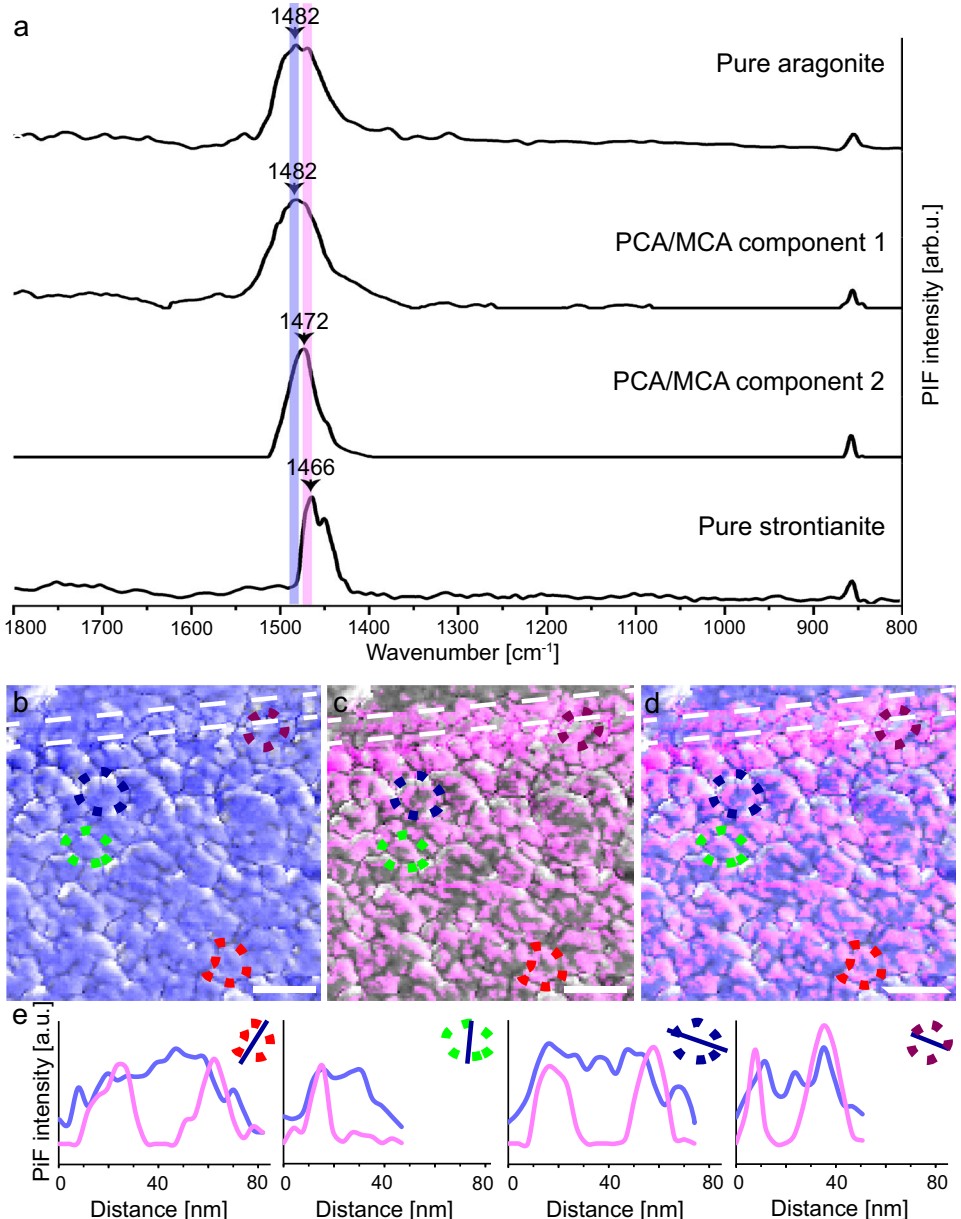

**Fig. 6 | Principal component analysis and multivariate curve resolution (PCA/MCR) applied to Sr-labelled nacre.** PCA/MCR was used to deconvolve the individual spectra underpinning the HyPIR map shown in Fig. 5. **a** Two pure component spectra were defined and corresponding maps showing the regions where these pure spectra are most prevalent were extracted. The PCA/MCR component spectra show their maximum intensities at different wavenumbers, namely at 1482 cm⁻¹ and at 1472 cm⁻¹ that correspond to aragonite and Sr-rich aragonite, respectively. The components are presented together with PiFM spot measurements of geological aragonite and synthetic strontianite reference materials for comparison (see method for details). The peak at 1472 cm⁻¹, defined as component 2, has an intermediate position between the main bands of aragonite and strontianite that suggest this component consists of a strontianite and aragonite mechanical mixture or

an aragonite-strontianite solid solutions beyond the 5 nm spatial resolution of the PiFM. The resulting maps exhibit within-granule heterogeneity of separate Sr-poor and Sr-rich aragonite areas. All component maps are shown superimposed on the greyscale phase contrast image depicting the space-filling nanogranular texture. Sr-rich areas are more common along the exteriors of individual granules, but rarely form fully enclosed cortices: **b** PCA/MCR component 1 (blue), **c** PCA/MCR component 2 (magenta) and **d** composite image of PCA/MCR component 1 and 2. Comparable results were obtained in an adjacent area to the one shown here (Supplementary Fig. 13). **e** transects across four representative nanogranules showing the intensity profiles for component 1 (blue) and 2 (magenta). Scale bars are 100 nm.

crystallisation pathway of nacre and represent phase separations combined with stabilisation of ACC by trace metal ions and/or organic molecules expelled upon crystallisation of aragonite[48]. Similarly, we interpret the Sr heterogeneity observed here as a result of phase transformation of ACC confined within nanogranules that are caused by the phase separation and the exclusion of excess Sr upon crystallisation of aragonite.

During aquaculture, Sr is non-selectively incorporated from the Sr-enriched seawater into the ACC phase[49] of the growing shell due to

the lack of long-range order of the ACC[15,50–52]. Subsequent crystallisation of ACC in the nanogranules is most likely triggered by dehydration[53] aided by the fact that the organic sheath, which delineates the nanogranular biomineralization compartments, is highly permeable for water[54]. Upon crystallisation in the confinement of nanogranules, phase separation manifests in areas of pure aragonite, separated from areas of strontianite or perhaps aragonite-strontianite solid solutions[55,56] beyond the 5 nm spatial resolution of PiFM. These Sr-rich areas are confined to individual nanogranules rather than

straddling their boundaries, as the nanogranules are individually coated by an organic sheath that prevents effective material transport. Hence, each organic-coated nanogranule[8,34,36,37] serves as a compartment for the crystallisation of aragonite from ACC in confinement[57].

The systematic presence of Sr-enriched and Sr-poor areas observed within individual nanogranules throughout the nacre tablet is a critical indication that the major transformation mechanism is the dissolution of the ACC to crystallise as aragonite. Upon transformation via a solid-solid mechanism, the chemical composition would remain unchanged and would therefore not be heterogeneous[58]. Dissolution and reprecipitation, on the other hand, involves material transport, thus creating heterogeneity, and leads to the heterogeneous distribution of aragonite and strontianite at the nanoscale. Dissolution and reprecipitation transformation from ACC to aragonite confined to individual nanogranules explain the preservation of the space-filling nanogranular texture of nacre. Spatially coupled dissolution-reprecipitation reactions that preserve the fine structures and overall shape of the replaced mineral, kinetically outcompete solid-solid transformation reactions and are, in fact, very common in mineralised systems[59]. They are catalysed by a fluid and are usually driven by small free energy differences between the reaction partner phases. Specifically, in interface-coupled systems with an interfacial fluid[58], the dissolution rate of one and the activation energy barrier for nucleation of the other reaction partner create local equilibrium conditions, which result in the preservation of very fine structural details after phase transformation[60,61].

Compared to natural biomineralization systems, a common issue encountered in laboratory dissolution-reprecipitation experiments is the significant morphological change occurring during transformation[14,15,18]. These previous experiments thus fail to explain the preservation of the hierarchical ultrastructure observed in natural biominerals and, thus, seem to support solid-solid transformation of ACC to the crystalline calcium carbonate phase. We provide here direct observational evidence in naturally formed nacre that the ACC-to-aragonite transformation proceeds via an intragranular and spatially confined dissolution-reprecipitation mechanism. Our findings resolve the long-standing question around the transformation processes of natural nacre and show that the fine structural details of the material are preserved by a dissolution-reprecipitation mechanism that takes place in confined compartments of organically coated nanogranules.

## Methods

### Aquaculture
Juvenile Mediterranean mussels *Mytilus galloprovincialis* (Lamarck, 1819) were collected alive from Twofold Bay, New South Wales, Australia and brought to the Macquarie University Seawater Facility. About 50 juvenile mussels sized 5–30 mm were evenly divided into two 50 L polyethylene tanks that were connected to a recirculating system for sterilised, filtered natural seawater sourced outside the Sydney harbour. The tanks were maintained at local ocean temperature, salinity and pH, while the lighting was adjusted to a circadian day/night cycle. The mussels were fed daily using a commercially available microalgae mix (Shellfish Diet 1800, Reed Mariculture Inc., USA)[21,31]. After 3 weeks of acclimatisation, the experimental period consisted of 35–55 days in which bivalves were transferred four times (group 1) and three times (group 2) from ambient seawater (8 μg/g Sr at a salinity of 35 psu) to conditions with elevated Sr concentrations of 120 μg/g (15× mean ocean water). The Sr-enriched seawater was produced by dissolving 0.37 g strontium dichloride hexahydrate (Merck KGaA, Germany) per litre into the tanks. Sr-labelling intervals were maintained for 3 days (group 1) and 6 days (group 2) and Sr-enriched seawater was fully replaced every 48 h. Between the labelling events, mussels were transferred back to ambient conditions for 6 (group 1) and 12 (group 2) days. After the last labelling event, mussels were collected and deep-frozen at −20 °C. After the experiments, soft tissues were removed,

shells were rinsed in deionized water and air-dried. Strontium was chosen as an elemental marker for pulse-chase labelling experiments, as Sr/Ca ratios in ACC are linearly dependent to the ratio in seawater[49].

### Sample preparation
Shells were mounted in EpoFix epoxy resin (Struers, Australia) and left to cure in the fridge at 2 °C. Lapping and polishing consisted of a series of 500–4000 grit diamond lapping discs, followed by 3 and 1 μm diamond suspensions and a diluted suspension of 0.04 μm colloidal silica (OP-S NonDry, Struers, Australia) concluded the polishing[21,62]. Polished mounts were used for PiFM without any additional preparation steps, while SEM-EDS and NanoSIMS analyses required sputter coating with carbon (20 nm) and gold (10 nm), respectively.

Specimens for APT were prepared using the Xe beam of a ThermoFisher Helios Hydra G4 UXe DualBeam Plasma Focussed Ion Beam-Scanning Electron Microscope (FIB-SEM) at the Australian Centre for Microscopy and Microanalysis (ACMM), University of Sydney. Regions of interest in the shells were BSE imaged to visualise the Sr-labelled nacre layers while accurately depositing the platinum protection strips onto the target areas for liftout (Supplementary Fig. 4). This approach avoided S/TEM imaging of the APT tips after milling, which can cause damage to the structure of the biomineral architecture, thus increasing the risk of fracturing during the APT run[31]. In situ liftouts were performed by using an accelerating voltage of 30 KV and a beam current of 1 nA. Annular milling was performed using a Zeiss Auriga FIB-SEM starting at 30 kV and 1 nA and stepping down the voltage and current to 10 kV and 50 pA. As nacre is a highly beam-sensitive material, it was crucial to use suitable FIB parameters and to maintain adequately low (i.e., <2 kV) electron-beam conditions for imaging during preparation[31].

### Scanning electron microscopy (SEM) and energy dispersive X-ray spectroscopy (EDS)
A Zeiss EVO MA15 tungsten SEM equipped with an Oxford 20 mm² X-max SDD EDS detector at Macquarie University was used to obtain quantitative SEM-EDS maps of Sr-labelled nacre. SEM-EDS mapping was carried out in high vacuum using the AZtec acquisition software (version 3.1) and operating conditions were 20 kV accelerating voltage, 12 mm working distance and 200 pA beam current. The primary electron beam was focused to a spot size of 400 nm that scanned a map area of 80 × 60 μm at a resolution of 1027 × 768 pixels. A ZAF correction was applied automatically by the software and major element concentrations resemble previous wavelength dispersive analysis[21]. Data processing was also performed using the AZtec 3.1 software.

High-resolution backscattered electron (BSE) images of polished cross-sections were taken using a JEOL JSM-7100F field emission gun-scanning electron microscope (FEG-SEM) at Macquarie University. Images were acquired at 15 kV acceleration voltage and 8 nA beam current.

### Nanoscale secondary ion mass spectrometry (NanoSIMS)
$^{88}Sr/^{40}Ca$ isotope abundances were mapped using a new generation CAMECA NanoSIMS 50 L ion probe at the University of Western Australia. Operation conditions followed published procedures[21]. A Hyperion RF plasma oxygen ion source produced a primary oxygen ion beam that was focused to a spot size of 100 nm to scan 50 × 50 μm map areas at a resolution of 512 × 512 pixels. Both Ca and Sr isotopes were measured on electron multipliers at 5000 mass resolution. Data reduction was performed using the OpenMIMS plugin (Harvard University) for ImageJ.

### Atom probe tomography (APT)
APT experiments were carried out on a CAMECA LEAP 4000Si at the ACMM, University of Sydney. This system is equipped with an ultraviolet laser that produces picosecond-pulses at an excitation wavelength of 355 nm. Operating parameters followed the protocol in Eder

et al.[31] and consisted of a pulse energy of 100–200 pJ, a pulse frequency of 160 kHz and a 0.5% detection rate that yielded a data set of 3 million detector hits (Supplementary Fig. 5). During analysis, the sample was cooled to 50 K in ultrahigh vacuum (<2 × 10⁻¹¹ Pa). The tip shape geometries of all datasets were calibrated by using SEM images, which gives a more accurate result for biominerals than using the voltage evolution curve[31]. Data reconstruction, 3D visualisation and compositional profiles were obtained using the IVAS software version v3.8.4. All compositional profiles (i.e., proxigrams, Fig. 4, Supplementary Figs. 7 and 9) were prepared from a cylindrical region (e.g., sized 48 × 48 × 31 nm in Fig. 4) traversing the APT data sub-sets. The data underpinning the compositional profiles were smoothed by a factor of 5 in Microsoft Excel and plotted in OriginPro. For a full list of the ion species detected in nacre APT data see Supplementary Table 1 in Eder et al.[31]. The uneven mineral-organic interface as well as differences in evaporation fields between the calcium carbonate and the organic matrices can lead to deviations in thickness and composition of the reconstructed data[31,63]. Hence, the APT data of this study is only used qualitatively.

We divided the line data shown in the compositional profiles into three sections referred to as Sr-poor aragonite, Sr-rich aragonite and interlamellar sheet (black dashed lines): The dashed line marking the transition from Sr-poor to Sr-rich aragonite areas was defined by a significant increase in Sr, while the dashed line marking the transition from the Sr-rich aragonite to the organic interlamellar sheet was set to the intercept point of Ca and C (Fig. 4, Supplementary Figs. 7 and 9).

### Photo-induced force microscopy (PiFM)

PiFM analyses were conducted using a VistaOne system at Molecular Vista Inc. San Jose, CA-USA. The method is based on conventional atomic force microscopy (AFM) and utilises the oscillating movements of a piezoelectric metal-coated cantilever while probing the sample surface. A wavelength-tunable IR laser is used to illuminate the sample area below the cantilever tip and the absorption in characteristic FTIR fingerprint regions is recorded by the increased attraction force on the cantilever[42]. We used a quantum cascade laser (QCL)-module from Block Engineering LLC as excitation source consisting of four serially connected gap-free QCLs that provide access to a wide wavenumber range of 770–1860 cm⁻¹. Measurements were conducted in dynamic AC AFM mode using a Nanosensors™ 300 kHz platinum iridium-coated, non-contact, silicon-based AFM probe (NanoWorld AG, Switzerland). For more details on the instrumentation and methodology see Otter et al.[42]. All PiFM data were processed in SurfaceWorks 3.0 Release 30. PiFM spectra were normalised to the laser power profile and smoothed with a factor of 17.

The absorption bands produced by PiFM correlate with those from traditional FTIR apart from the fact that PiFM bands are narrower, due to the monochromatic excitation source and the significantly smaller sampling area[43]. Hence, PiFM can resolve nanoscale phase heterogeneity more appropriately and even separate individual neighbouring peaks that are not spectrally resolved in the broader FTIR absorption band envelopes. For this reason, we include PiFM absorption spectra of geological aragonite (microanalytical reference material VS001/1-A[64]) and synthetic strontianite (BDH Laboratory Reagents, Dubai, United Arab Emirates) for comparison (Fig. 6a). These spectra were obtained using a VistaOne system at the ANU Research School of Earth Sciences following the analytical conditions outlined above.

### Principal component analysis (PCA) and multivariate curve resolution

PCA/MCR was performed in SurfaceWorks 3.0 Release 30 using the PCA/MCR plugin. All input spectra from the HyPIR map shown in Fig. 5 and Supplementary Fig. 12 were divided by the power profile of the laser to remove any convolution contributions from the QCL laser and

then smoothed using a Savitzky–Golay function. A set of four loading vectors and score maps were used to extract component 1 and 2 (shown in Fig. 6a, b, Supplementary Fig. 13) from the remaining organic phases and to reduce noise. In the last step, the final component maps were saved as separate data channels and exported to ImageJ version 2.9.0 for further processing, while component spectra were exported and further processed in Microsoft Excel.

### Statistics and reproducibility

For a list of samples and experiments see Supplementary Table 1.

### Reporting summary

Further information on research design is available in the Nature Portfolio Reporting Summary linked to this article.

## Data availability

All relevant data supporting the findings of this study are provided in the Supplementary Information file and are available from the corresponding author upon request.

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

## Acknowledgements

We thank Eden Mussels for providing *M. galloprovincialis*. We acknowl-
edge Assoc. Prof Jane Williamson and Josh Aldridge for assistance at the
Macquarie Seawater Facility and we are grateful to Dr. Wayne O'Connor
(Port Stephens Fisheries Centre, NSW Department of Primary Industries)
for insightful discussions on bivalve aquaculture. We thank Dr. Paul
Guagliardo for his assistance with NanoSIMS analysis and Dr. Sean Murray
for his assistance with SEM imaging, Dr. Timothy D. Murphy for assistance
with quantitative SEM-EDS mapping and Dr. Oscar Branson for discus-
sions on APT data. The authors are grateful for the scientific and technical
input and support from Microscopy Australia at the University of Sydney
and the University of Western Australia. We acknowledge the Macquarie
University Faculty of Science and Engineering Microscope Facility
(MQFoSE MF) for access to its instrumentation and support from Sue
Lindsay. This study is supported by the Australian Research Council
(DP160102081 and DP210101268 to D.E.J.) and a Beate Mocek Price of the
German Mineralogical Society awarded to L.M.O.

## Author contributions

L.M.O. and D.E.J. designed the study. L.M.O. performed aquaculture
experiments, prepared samples and participated in data collection. K.E.
and J.M.C. conducted atom probe experiments. L.Y. prepared APT tips.
M.R.K. performed NanoSIMS analysis. D.B.N. and P.O'.R. performed PiFM
analysis. L.M.O. wrote the manuscript with contributions from all co-
authors.

## Competing interests

The authors declare no competing interests.

## Additional information

**Supplementary information** The online version contains
supplementary material available at

L. M. Otter.

**Peer review information** *Nature Communications* thanks the anon-
ymous reviewers for their contribution to the peer review of this
work. Peer reviewer reports are available.

