## [Peer Review File · Nature Communications]

Growth dynamics and amorphous-to-crystalline phase transformation in natural nacreReviewers' Comments:

Reviewer #1:

Remarks to the Author:

Comments on the manuscript "Growth dynamics and amorphous-to-crystalline phase transformation in natural nacre" submitted by Oter et al.

Abstract:

"Most marine calcifiers form their shells from amorphous calcium carbonate by particle attachment and stepwise crystallisation of metastable precursor phases".

This is not a fact, but a theory still under discussion.

Introduction:

"In fact, crystallization via particle attachment and stepwise transformation of metastable precursor phases are now recognized as crystallization mechanisms for many different systems".

As said above, particle attachment is a theory. The presence of metastable material is usually admitted (Gal, Weiss, Nudelman, Addadi....). They are not in the references.

"Amorphous calcium carbonate (ACC), the first solid phase formed in bio-carbonates, incorporates elements indiscriminately^{12,13}".

I do not find such assertions in the cited references 12, 13.

Does not show chemical data, 13 is about model and do not take into account the organic contents; thus a comparison between results of modeling and natural biomineral is not relevant or the chemistry of the shells.

"Correlative analysis using Nanoscale Secondary Ion Mass Spectrometry, portions for the first time down to the atomic scale."

Usually, "correlative analysis" is used when several characteristics of a sample are simultaneously obtained in a single run using a single analytical system. Here, data were acquired using different systems.

"We show that nacre forms via a two-stepped process of extensional followed by space-filling growth, a process that has long been suggested but is observed here directly for the first time."

References are missing.

"creating a characteristic nanogranular structure³⁰⁻³³."

Presence of nanogranules in mollusk shells has been suspected using TEM and SEM (Mutvei 1971). Then, it was shown in aragonite and calcite has been evidenced by Dauphin (2001) using AFM. A thin organic cortex is also shown. Then it was said that the cortex surrounded granules are a mixture of organic and ACC. Colour changes within a granule seen in phase image contrast reveal that the inner content of a granule is not homogeneous. Similar granules were also described by Addadi et al.

"For the three PiFM phase distribution maps (Figs. 5b-d) obtained simultaneously with the topography map, aragonite was mapped using the characteristic band at 1470 cm⁻¹ (Fig. 5b, e 42). Strontianite (Fig. 5c, e; green) was mapped via its peak associated with the asymmetric stretching mode that was visible as a small shoulder on one side of the aragonite band (Supplementary Fig. 9) at 1444 cm⁻¹ 42."

At this stage, PiFM is not explain. In Material and methods, it is not said what are the kind of data obtained with this system. I am somewhat familiar with infrared spectra, so I suspect that IR data are obtained, but I have to look for the website of the provider to be sure.

Ref 42 about high temperature minerals is not appropriate for shells. 1444 is not a strontianite band (Jones & Jackson).

"For example, the organic interlamellar sheet between the nacre tablets is dominated by the organic moiety (red)."

I do not understand this sentence. What is the other moiety?

"Further, high intensities for strontianite (green) are clearly seen to be enriched adjacent to the organic interlamellar sheet and particularly in the cortices of nanogranules in this region."

Two problems:

- granules are not well displayed in figures, and cortex is not visible.
- presence of strontianite is not demonstrated in this manuscript.

See also Finch et Allison.

"Cortices of similar dimensions in nacre, including in nacre nanogranules have been previously reported in the literature,"

References ?

Reviewer #2:

Remarks to the Author:

Minor comments:

- (1) The authors noted that the nanograins adjacent to the organic sheet are more enriched in Sr compared to other areas in the nacre tablets but do not explain this phenomenon.
- (2) Since the crystallization via particle attachment is proposed as a common biocrystallization mechanism, and the occurrence of the nanograins outlined by some organic-enriched "envelopes" was commonly observed, it would be worthy to provide a short perspective passage in which similar mechanism as observed in nacre would be suggested for similarly textured biominerals.
- (3) Could the authors propose some explanation of discrepancy [?] between proposed mechanism of crystallization and removal of [excess of strontium] organic components along the outer contours of the nanograins (ca. 20-150 nm in diameter) vs. organic inclusions observed within nacre lamellae (Younis et al. 2012 Cryst Growth Des 12:4574) which do not delineate individual nanograins (which in fact are not observed in TEM)? Such [?] discrepancy was also noted between AFM and TEM observations of aragonite fibers of corals (e.g., Benzerara et al. Ultramicroscopy 111(8):1268-1275).

We appreciate the constructive comments from both reviewers and thank them for their feedback about specific aspects of our work. Based on their feedback, we have made several substantial changes to the manuscript and the figures as outlined below (blue font). We appreciate that the manuscript has significantly improved due to the suggestions of the referees.

Reviewer 1:

1. Abstract: “Most marine calcifiers form their shells from amorphous calcium carbonate by particle attachment and stepwise crystallisation of metastable precursor phases”. This is not a fact, but a theory still under discussion.

The sentence now reads: “Most marine calcifiers form their shells from amorphous calcium carbonate, hypothesized to occur via particle attachment and stepwise crystallisation of metastable precursor phases.”

2. Introduction: “In fact, crystallization via particle attachment and stepwise transformation of metastable precursor phases are now recognized as crystallization mechanisms for many different systems”. As said above, particle attachment is a theory. The presence of metastable material is usually admitted (Gal, Weiss, Nudelman, Addadi....). They are not in the references.

We edited the sentence accordingly and it now reads: “In fact, crystallization via particle attachment and stepwise transformation of metastable precursor phases are now accepted as crystallization models for many different systems^{1,6,7}.” Further, we added citations to Gal et al. (2014) and Nudelman (2015). References to Weiss et al. (2002) and Addadi et al. (2003) are already cited (as ref #5 and #1 in the submitted manuscript version).

3. “Amorphous calcium carbonate (ACC), the first solid phase formed in bio-carbonates, incorporates elements indiscriminately^{12,13}”. I do not find such assertions in the cited references 12, 13. Does not show chemical data, 13 is about model and do not take into account the organic contents; thus a comparison between results of modelling and natural biomineral is not relevant or the chemistry of the shells.

We removed the citations and rewrote this part of the manuscript: “As a general principle, amorphous materials lack defined partition coefficients and are thus less selective with respect to trace element incorporation compared to their crystalline counterparts.”

4. “Correlative analysis using Nanoscale Secondary Ion Mass Spectrometry, portions for the first time down to the atomic scale.” Usually, “correlative analysis” is used when several characteristics of a sample are simultaneously obtained in a single run using a single analytical system. Here, data were acquired using different systems.

We replaced “correlative” with “correlated” throughout the manuscript.

5. “We show that nacre forms via a two-stepped process of extensional followed by space-filling growth, a process that has long been suggested but is observed here directly for the first time.” References are missing.

We added citations to Bevelander, G. & Nakahara (1969), Nakahara, H. & Bevelander (1971), and Addadi, L. & Weiner (1971).

6. “creating a characteristic nanogranular structure 30–33.” Presence of nanogranules in mollusk shells has been suspected using TEM and SEM (Mutvei 1971). Then, it was shown in aragonite and calcite has been evidenced by Dauphin (2001) using AFM. A thin organic cortex is also shown. Then it was said that the cortex surrounded granules are a mixture of organic and ACC. Colour changes within a granule seen in phase image contrast reveal that the inner content of a granule is not homogeneous. Similar granules were also described by Addadi et al.

We added a citation to Dauphin (2008), which we think emphasises the presence of organic cortices in more detail.

7. “For the three PiFM phase distribution maps (Figs. 5b-d) obtained simultaneously with the topography map, aragonite was mapped using the characteristic band at 1470 cm⁻¹ (Fig. 5b, e 42). Strontianite (Fig. 5c, e; green) was mapped via its peak associated with the asymmetric stretching mode that was visible as a small shoulder on one side of the aragonite band (Supplementary Fig. 9) at 1444 cm⁻¹ 42.” At this stage, PiFM is not explain. In Material and methods, it is not said what are the kind of data obtained with this system. I am somewhat familiar with infrared spectra, so I suspect that IR data are obtained, but I have to look for the website of the provider to be sure.

We changed this part of the results to explain and clarify the relatively new method. It now reads: “PiFM is a nanoscale AFM-based technique that raster-scans the sample surface at high spatial resolution. It measures the force created in a very small interaction volume between the AFM cantilever tip and the sample surface, while laser sweeps through the infrared spectrum causing absorption-specific variations in the recorded force⁴². PiFM absorption spectra agree with those produced by traditional FTIR spectroscopy⁴³ but are obtained at a higher spectral and spatial resolution of few nanometres per spot⁴², which can lead to some differences in absorption band appearance⁴⁴ particularly for materials that are heterogeneous at the nanoscale” In addition, we have also rewritten the method section on PiFM for clarification.

8. Ref 42 about high temperature minerals is not appropriate for shells. 1444 is not a strontianite band (Jones & Jackson).

We removed the reference and now cite Jones and Jackson. In addition, we have included PiFM reference spectra from pure aragonite and strontianite for comparison that serve as endmembers for the phases found in the shells. The reviewer is correct in stating that the band positions are not entirely straight forward to be assigned to strontianite. The reason for this is that it is a mixture of aragonite and strontianite. The main strontianite band at 1466 cm⁻¹ is interfered by the aragonite main band and we rely on the smaller strontianite band at 1446 cm⁻¹ whenever showing spectra that were not deconvoluted. In the revised manuscript, we have carefully verified the position of this smaller band by analysis of a synthetic, lab-grade strontianite and now rely on our own measurements for the band positions for consistency rather than on the literature. In contrast to traditional FTIR, the smaller peak at 1446 cm⁻¹ is detectable in the PiFM spectra here as PiFM has higher resolution (Benedis et al., 2022).

In the revised version of the manuscript, we provide Voigt deconvolution models for PiFM spectra of the strontium labelled nacre and of the synthetic strontianite reference material (new Supplementary Figure 11). In addition, we provide a new Figure 6 showing Principal Component Analysis and Multivariate Curve Resolution (PCA/MCR) for the mapped area in Figure 5. The resulting component maps (new Figure 6b) visualize the heterogeneous distribution of Sr-rich aragonite and the Sr-poor aragonite. This is now clarified in the manuscript.

9. “For example, the organic interlamellar sheet between the nacre tablets is dominated by the organic moiety (red).” I do not understand this sentence. What is the other moiety?

We have rewritten this part for clarification.

10. “Further, high intensities for strontianite (green) are clearly seen to be enriched adjacent to the organic interlamellar sheet and particularly in the cortices of nanogranules in this region.” Two problems: granules are not well displayed in figures, and cortex is not visible. presence of strontianite is not demonstrated in this manuscript. See also Finch et Allison.

We have modified the presentation of the 1 x 1 μm map from the submitted version with two cropped map sections sized 0.5 x 0.5 μm maps that show the nanogranular texture and the heterogeneous phase distribution clearly. We have replaced the term “cortex” with “heterogeneity” throughout the manuscript, which we find more appropriate. As discussed above, we now demonstrate the assignment of the 1446 cm⁻¹ peak to strontianite.

10. “Cortices of similar dimensions in nacre, including in nacre nanogranules have been previously reported in the literature,” References ?

We now reference Nassif et al. (2005) and Macias-Sanchez et al. (2017) and have rewritten this part of the manuscript.

Reviewer 2:

1. Minor comments: The authors noted that the nanograins adjacent to the organic sheet are more enriched in Sr compared to other areas in the nacre tablets but do not explain this phenomenon.

Unfortunately, this map is not an intensity map but is a result of an earlier attempt at Principal Component Analysis/ Multivariate Curve Resolution (PCA/MCR) which was included erroneously. We apologize for this mistake and have now included the correct map in Fig. 5. In addition, we have replaced the composite map in Fig. 6 with new PCA/MCR maps. We found that the apparent higher intensity along the organic sheet is an artefact of the PCA/MCR that arises from scaling intensities of the peaks in this area of the map. The raw data indeed shows peaks in this area to be higher in intensity in addition to some peak broadening adjacent to the organic-rich area:

Representative raw HyPIR spectra illustrating variations in peak shape throughout the map: red line sampled adjacent to the organic sheet showing a higher signal and green sampled further away from the sheet.

2. Since the crystallization via particle attachment is proposed as a common biocrystallization mechanism, and the occurrence of the nanograins outlined by some organic-enriched "envelopes" was commonly observed, it would be worthy to provide a short perspective passage in which similar mechanism as observed in nacre would be suggested for similarly textured biominerals.

In response to reviewer 1 we have now toned down the evidence of particle attachment as a general process in biomineralization. While the existence of space-filling nanogranular texture is indeed wide-spread in mineralized systems this does not automatically serve as evidence for crystallization via particle attachment. To our knowledge, only nacreous bivalves have been shown to form via nanogranules that are coated by a pronounced organic sheath.

3. Could the authors propose some explanation of discrepancy [?] between proposed mechanism of crystallization and removal of [excess of strontium] organic components along the outer contours of the nanograins (ca. 20-150 nm in diameter) vs. organic inclusions observed within nacre lamellae (Younis et al. 2012 Cryst Growth Des 12:4574) which do not delineate individual nanograins (which in fact are not observed in TEM)? Such [?] discrepancy was also noted between AFM and TEM observations of aragonite fibers of corals (e.g., Benzerara et al. Ultramicroscopy 111(8):1268-1275).

The revised version of the manuscript now contains the correct PiFM map with the artefact removed that erroneously suggested Sr enrichment along the organic interlamellar sheet delineating the nacre tablet. We do not suggest 'removal of [excess of strontium] organic components' and are not entirely sure how the reviewer would gain that impression. Hopefully the revised version of the manuscript has clarified any potential misrepresentations.

We understand the reviewer's comment as asking us to address the nature and distribution of organic inclusions in nacre and the differences of variable analytical methods in depicting them accurately. Firstly, organic inclusions in nacre are regularly visualized by SEM and TEM in the literature, and have, in fact, been shown in our recent work

to be important for the mechanical properties of nacre (Gim et al., 2019). They are recognized via different z-contrasts in SEM and S/TEM which imply less dense elemental components. Whether these inclusions in nacre are fully organic or whether they represent a mixture of organics plus mineral plus water (bivalve shells contain several wt.% of water) is yet unresolved but equally possible. Nacreous bivalve shells contain ca. 4% of organics, which is considerably more than corals, where the reviewer mentions that TEM fails to detect the organic sheaths coating individual nanogranules and any organic inclusions (Benzerara et al., 2011). This is however not the case in nacreous bivalve shells, where organic coatings can be seen throughout by S/TEM analysis (Eder et al., 2019; Hovden et al., 2015; Jacob et al., 2008; Jacob et al., 2011). Why the thin organic coatings do not show up prominently in HRTEM analysis is perhaps a question more appropriately addressed at a TEM specialist.

Our study is targeted towards the mineral content of nacre and not on organic inclusions, hence we chose areas with minimal organic inclusions. We were able to visualise organic inclusions in nacre with PiFM previously (Otter et al., 2021) but we agree with the reviewer that it depends on the analytical method chosen as well as on the sample preparation whether these inclusions and thin organic coatings can be visualized. They are clearly more challenging to visualize in traditional AFM modes, perhaps because this is a surface method and uses highly polished surfaces. As we rated the visualization of organic inclusions as not of primary interest to this study here, we feel that commenting on the complex topic of sample preparation and specific advantages and disadvantages of analytical approaches to be beyond the focus of this study.

References

- Benedis, D.V., Dazzi, A., Rivallan, M., Pirngruber, G.D., 2022. Surface Heterogeneity in Amorphous Silica Nanoparticles Evidenced from Tapping AFM-IR Nanospectroscopy. *Analytical Chemistry*.
- Dauphin, Y., 2008. The nanostructural unity of mollusc shells. *Mineralogical Magazine* 72 (1), 243–246.
- Eder, K., Otter, L.M., Yang, L., Jacob, D.E., Cairney, J.M., 2019. Overcoming Challenges Associated with the Analysis of Nacre by Atom Probe Tomography. *Geostandards and Geoanalytical Research* 43 (3), 385–395.
- Gim, J., Schnitzer, N., Otter, L.M., Cui, Y., Motreuil, S., Marin, F., Wolf, S.E., Jacob, D.E., Misra, A., Hovden, R., 2019. Nanoscale deformation mechanics reveal resilience in nacre of *Pinna nobilis* shell. *Nature Communications* 10 (1), 1–8.
- Hovden, R., Wolf, S.E., Holtz, M.E., Marin, F., Muller, D.A., Estroff, L.A., 2015. Nanoscale assembly processes revealed in the nacropismatic transition zone of *Pinna nobilis* mollusc shells. *Nature Communications* 6 (10097), 1–7.
- Jacob, D.E., Soldati, A.L., Wirth, R., Huth, J., Wehrmeister, U., Hofmeister, W., 2008. Nanostructure, composition and mechanisms of bivalve shell growth. *Geochimica et Cosmochimica Acta* 72 (22), 5401–5415.
- Jacob, D.E., Wirth, R., Soldati, A.L., Wehrmeister, U., Schreiber, A., 2011. Amorphous calcium carbonate in the shells of adult Unionoida. *Journal of Structural Biology* 173 (2), 241–249.
- Otter, L.M., Förster, M.W., Belousova, E., O'Reilly, P., Nowak, D., Park, S., Clark, S., Foley, S.F., Jacob, D.E., 2021. Nanoscale Chemical Imaging by Photo-Induced Force Microscopy: Technical Aspects and Application to the Geosciences. *Geostandards and Geoanalytical Research* 45 (1), 5–27.

Reviewers' Comments:

Reviewer #1:

Remarks to the Author:

The revised manuscript is improved for text, references and illustrations. Changes are made following the comments of the reviewer.

In my opinion, the new version is ready to be published.

Reviewer #2:

Remarks to the Author:

I am very pleased with replies of the authors, especially for: (1) correction of the map in Fig. 5, (2) providing broader context of existence of nanogranular textures, (3) commenting on nature of nanograins and possible organic "envelopes"

This is excellent contribution - ready to be published in my opinion.

We appreciate the feedback from both reviewers and thank them for their time and effort, which has helped us improve our manuscript.

Reviewer 1:

The revised manuscript is improved for text, references and illustrations. Changes are made following the comments of the reviewer. In my opinion, the new version is ready to be published.

We thank this reviewer for having another thorough look at our manuscript and for their positive comments.

Reviewer 2:

I am very pleased with replies of the authors, especially for: (1) correction of the map in Fig. 5, (2) providing broader context of existence of nanogranular textures, (3) commenting on nature of nanograins and possible organic "envelopes". This is excellent contribution - ready to be published in my opinion.

We thank this reviewer for re-evaluating our manuscript and their support.